# Multilocus Genetic Profile Reflecting Low Dopaminergic Signaling Is Directly Associated with Obesity and Cardiometabolic Disorders Due to Antipsychotic Treatment

**DOI:** 10.3390/pharmaceutics15082134

**Published:** 2023-08-14

**Authors:** Aurora Arrue, Olga Olivas, Leire Erkoreka, Francisco Jose Alvarez, Ainara Arnaiz, Noemi Varela, Ainhoa Bilbao, Jose-Julio Rodríguez, María Teresa Moreno-Calle, Estibaliz Gordo, Elena Marín, Javier Garcia-Cano, Estela Saez, Miguel Ángel Gonzalez-Torres, Mercedes Zumárraga, Nieves Basterreche

**Affiliations:** 1Mental Health Network Group, BioCruces Bizkaia Health Research Institute, 48903 Barakaldo, Spain; olga.olivasgallego@osakidetza.eus (O.O.); leire.erkorekagonzalez@osakidetza.eus (L.E.);; 2Neurochemical Research Unit, Bizkaia Mental Health Network, Osakidetza Basque Health Service, 48903 Barakaldo, Spain; 3Bizkaia Mental Health Network, Zaldibar Hospital, Osakidetza Basque Health Service, 48250 Zaldibar, Spain; 4Department of Psychiatry, Galdakao-Usánsolo University Hospital, Osakidetza Basque Health Service, 48960 Galdakao, Spain; 5Department of Neurosciences, Faculty of Medicine and Dentistry, University of the Basque Country UPV/EHU, 48940 Leioa, Spain; 6Research Unit, Cruces University Hospital, Osakidetza Basque Health Service, 48903 Barakaldo, Spain; 7Erandio Mental Health Center, Bizkaia Mental Health Network, Osakidetza Basque Health Service, 48950 Erandio, Spain; 8Uribe Mental Health Center, Bizkaia Mental Health Network, Osakidetza Basque Health Service, 48990 Getxo, Spain; 9Animal Research Facility, BioCruces Bizkaia Health Research Institute, 48903 Barakaldo, Spain; 10Functional Neuroanatomy, BioCruces Bizkaia Health Research Institute, Ikerbasque Basque Foundation for Science, 48903 Barakaldo, Spain; 11Bizkaia Mental Health Network, Zamudio Hospital, Osakidetza Basque Health Service, 48170 Zamudio, Spain; 12Bizkaia Mental Health Network, Bermeo Hospital, Osakidetza Basque Health Service, 48370 Bermeo, Spain; 13Alternatives to Hospitalization in Bilbao, Bizkaia Mental Health Network, Osakidetza Basque Health Service, 48903 Bilbao, Spain; 14Integrative Research Group in Mental Health, BioCruces Bizkaia Health Research Institute, 48013 Bilbao, Spain; 15Department of Psychiatry, Basurto University Hospital, Osakidetza Basque Health Service, 48013 Bilbao, Spain

**Keywords:** atypical antipsychotic, dopaminergic system, polymorphisms, multilocus genetic profile score, cardiometabolic disorders

## Abstract

Treatment with second-generation antipsychotics (SGAs) can cause obesity and other cardiometabolic disorders linked to D2 receptor (DRD2) and to genotypes affecting dopaminergic (DA) activity, within reward circuits. We explored the relationship of cardiometabolic alterations with single genetic polymorphisms *DRD2* rs1799732 (NG_008841.1:g.4750dup -> C), *DRD2* rs6277 (NG_008841.1:g.67543C>T), *COMT* rs4680 (NG_011526.1:g.27009G>A), and VNTR in both *DRD4* NC_000011.10 (637269-640706) and *DAT1* NC_000005.10 (1392794-1445440), as well as with a multilocus genetic profile score (MLGP). A total of 285 psychiatric patients treated with SGAs for at least three months were selected. Cardiometabolic parameters were classified according to ATP-III and WHO criteria. Blood samples were taken for routinely biochemical assays and PCR genotyping. Obesity (BMI, waist (W)), high diastolic blood pressure (DBP), and hypertriglyceridemia (HTG) were present in those genetic variants related to low dopaminergic activity: InsIns genotype in rs1799732 (BMI: OR: 2.91 [1.42–5.94]), *DRD4-*VNTR-L allele (W: OR: 1.73 [1.04–2.87]) and 9R9R variant in *DAT1*-VNTR (W: OR: 2.73 [1.16–6.40]; high DBP: OR: 3.33 [1.54–7.31]; HTG: OR: 4.38 [1.85–10.36]). A low MLGP score indicated a higher risk of suffering cardiometabolic disorders (BMI: OR: 1.23 [1.05–1.45]; W: OR: 1.18 [1.03–1.34]; high DBP: OR: 1.22 [1.06–1.41]; HTG: OR: 1.20 [1.04–1.39]). The MLGP score was more sensitive for detecting the risk of suffering these alterations. Low dopaminergic system function would contribute to increased obesity, BDP, and HTG following long-term SGA treatment.

## 1. Introduction

Patients with severe mental disorders, such as schizophrenia or bipolar disorder, receiving antipsychotic (AP) treatment have a higher risk of developing obesity, as well as metabolic and blood pressure disorders, than the general population. First-generation antipsychotics can produce obesity and other cardiometabolic alterations, and second-generation antipsychotics (SGAs) produce a 1.5-fold risk increase, with clozapine and olanzapine being the most harmful [1,2]. These obesity-related cardiometabolic disorder parameter alterations can lead to the development of metabolic syndrome after long-term SGA treatment, thus increasing the risk of endocrine and cardiovascular diseases [3], morbidity, and mortality [4]. Compared with the general population, in patients treated with antipsychotics, obesity prevalence is estimated as 45–55%, metabolic syndrome prevalence is estimated as 24–52%, and life expectancy is reduced by 20% [1,4,5].

Predisposition to suffer side effects with antipsychotic treatment varies between individuals, as well as their type and severity. Family genetic studies and genome-wide association studies have shown that obesity produced by antipsychotics is influenced by genetic and environmental factors [6,7]. Recent literature supports the participation of polymorphisms in different genes, common to the general population, related to systems targeted by antipsychotics, as well as hormones inducing satiety and energy [7,8]. However, there are no markers that identify those patients vulnerable to these side effects, which might guide professionals to treatment choice, selection, and application.

The underlying mechanisms involved in the development of obesity and metabolic alterations induced by the SGAs are multifactorial and remain unclear [9]. Direct effects have been described at both central and peripheral levels. At the central level, several studies relate the effects of SGAs such as D2 dopaminergic receptor (DRD2) antagonists to these negative effects [10,11,12,13,14], influencing the neuronal systems associated with food addiction and reward mechanisms in DA-dependent mesolimbic and mesocortical pathways [15,16].

It is theorized that a hypo-responsive reward circuit pushes overfeeding to compensate for the lack of DA [15,17,18]. In animal models with low DA levels, feeding-enhancing behavioral tasks, such as overeating, help to restore normal DA levels by engaging in reward-predicting activities [19]. This is in line with the fact that the administration of DA agonists reduces food intake, leading to weight loss [20,21], and DRD2 blockade increases food intake, resulting in weight gain [11,22,23]. Likewise, functional neuroimaging studies in humans have shown deficiencies in the dopaminergic reward pathways, demonstrating that a decrease in the availability of DRD2 and the DAD4 (DRD4) receptors is determinant for obesity.

Consistent with this hypothesis, genotypes associated with reduced DA signaling have been shown to correlate with obesity and lipid alterations [24,25,26,27,28,29,30], and to influence eating behavior by means of increased intake frequency and preference for high-fat foods [31,32]. In this sense, individuals with the InsIns genotype of the *DRD2*rs1799732 (-141C Ins/Del; NG_008841.1:g.4750dup -> C), and those with the TT genotype of *DRD2*rs6277 (NG_008841.1:g.67543C>T), both associated with a lower DRD2 density in the striatum [33,34]), have been described as being at higher risk for developing eating disorders and obesity compared with individuals with InsDel genotypes in *DRD2*rs1799732 and CT and CC genotypes in *DRD2*rs6277 [25,26,28]. In the *DRD4* gene, when studying polymorphism of variable numbers (48 base pairs) of tandem repeats (VNTR) (NC_000011.10 (637269-640706 insertion), individuals with the seven-repeat or longer allele (DRD4-L), which confers low DA signaling in the striatum [35,36], were shown to have a higher risk for obesity than individuals with shorter alleles (DRD4-S) [37,38]. In terms of VNTR polymorphism (40 base pairs) in the DAT gene (*DAT1*/*SLCA3:* NC_000005.10 (1392794-1445440), complement), individuals with the nine-repeat allele (9R), associated with increased DAT1 expression [39], correlate with an increased risk for obesity compared with individuals with ten-repeat allele (10R) [40,41]. Lastly, in the catechol O-methyltransferase gene (*COMT*), rs4680 (Val158Met), (NG_011526.1:g.27009G>A; G to A base change, resulting in a change of the amino acid valine (Val) to a methionine (Met) in the protein, which results in reduced enzyme activity); it has been shown that the GG (ValVal) genotype, related to a greater enzyme activity [42], is associated with obesity and hypercholesterolemia [43,44].

Evidence supporting weight gain and alterations in metabolic indicators after administration of SGAs is controversial. While associations between weight gain and metabolic traits have been found in low-DA-activity genotypes of SNPs in *DRD2* rs1799732 and rs6277 [45,46,47], VNTR in the *DRD4* gene [48] and in the *COMT* gene, and the ValVal rs4680 genotype [49], other studies are inconsistent with this hypothesis [46,49,50,51,52,53,54,55]. A single study evaluating the relationship between *DAT1* and weight gain produced by the administration of antipsychotics did not report an association [54].

There is extensive evidence implicating the influences of different common polymorphisms on multiple genes, each with a small effect on the development of disorders, behaviors, and complex diseases such as obesity [7,8]. Furthermore, since polygenic disorders are genetically heterogeneous, the associations described may not be replicated equally in all studies [56]. To overcome this, several groups have shown that the additive effect of multiple genes within a biological pathway exhibits greater predictive power than any single SNP within the score [41,57,58,59]. Thus, a biologically informed polygenic index, called “multilocus genetic profile” (MLGP) score, combining the influence of multiple polymorphic dopamine functional markers (polymorphisms in the genes *DRD2*, *DRD4*, *DAT1*, and *COMT*), was used to determine the genetic risk. All markers were already individually associated with variations in striatal dopamine signaling. These studies showed that the use of the MLPG score demonstrated greater reliability or influence on striatal dopaminergic signaling than each locus individually in healthy volunteers [41,57,58,59].

On the basis of the described findings, we hypothesized that those patients with a low DA activity genetic profile (low density or affinity for DRD2 and DRD4, higher DAT reuptake, and higher DA catabolism through COMT, which would contribute to a lower availability of DA) may have higher risk of obesity and adverse cardiometabolic effects when treated with SGAs. Thus, our first aim was to determine the associated risk for each individual genotype (*DRD2*rs1799732, *DRD2*rs6277, *DRD4*-VNTR, *DAT1*-VNTR, and *COMT*rs4680) in a population of patients receiving antipsychotic treatment. In addition, we analyzed whether the combined risk of the five genotypes (MLGP score) in this same cohort of patients resulted in a greater risk or sensitivity than the individual genotypes.

## 2. Materials and Methods

The sample consisted of 285 patients with severe mental disorders, according to DSM-IV criteria, who received SGA treatment for a minimum period of 3 months. All participants were Caucasian, of both sexes, and between 18 and 65 years of age. The patients came from the Basque Health Service, Osakidetza in Bizkaia, Spain. Patients with congenital endocrine-metabolic diseases, eating disorders, and substance use disorder, who were pregnant or lactating, and who habitually consumed substances of abuse were excluded. Occasional use of alcohol and cannabis was allowed (one or two days per week). All patients were informed of the details of the study and signed consent to participate in the study. The project was carried out following the requirements of the Declaration of Helsinki and with the approval of the Ethics Committee of the Basque Country. As this was a naturalistic study, there were no specific treatment guidelines (medications and/or psychotherapy).

The following variables were colle”ted:’age, sex, age of the psychiatry disorder onset, level of education, marital status, tobacco, alcohol and occasional cannabis use, blood pressure, AP treatment type and daily dose, concomitant treatments, and treatment for cardiometabolic alterations. The duration of the illness was calculated as the difference between the age of recruitment and psychiatric illness onset. The prescribed daily doses of Aps were converted to an estimated equivalent amount of chlorpromazine according to the international consensus [60].

Patients were assigned to the monotherapy group if they only received SGA and to the polytherapy group if, in addition to an SGA, they received more than one antipsychotic (SGA, typical or first-generation antipsychotic). In both groups, treatment with mood stabilizers, antidepressants, and anxiolytics was allowed.

Body mass index (BMI) was used to assess general obesity and waist circumference (W) to determine abdominal obesity. Weight and height parameters were collected for BMI calculation [weight (kg)/height (m)^2^]. An age-corrected BMI over 25 was considered an indicator of cardiometabolic risk, following WHO criteria. W was measured at the midpoint of the abdominal area, between the lower margin of the tenth rib and the upper edge of the iliac crest, after a normal expiratory breath. The criteria established by the National Cholesterol Education Program Adult Treatment Panel III (ATP III) was used to determine cardiometabolic risk according to the presence of central abdominal obesity (W: women > 88 cm, men > 102 cm).

Blood pressure data and serum triglyceride (TG), high-density lipoprotein cholesterol (HDL), and glucose levels were collected following ATP-III criteria: blood pressure: ≥130/85 mm Hg or treatment with antihypertensive drugs, TG levels: ≥150 mg/dL and HDL levels: women < 50 mg/dL, men < 40 mg/dL), treatment with lipid-lowering drugs and glucose levels: ≥100 mg/dL, or treatment with hypoglycemic drugs.

Patients were considered to have high blood pressure when they had altered levels of systolic (SBP) and diastolic blood pressure (DBP). Patients were also considered to have dyslipidemia when they presented altered serum levels of TG (hypertriglyceridemia, (HTG)), HDL, or both.

Samples for DNA collection and biochemical assays were obtained between 8 and 8:30 a.m., after 12 h of fasting. Serum fasting concentrations of TG, HDL, and glucose were assessed using standard techniques in laboratory analysis.

DNA was extracted from the blood samples via a commercial kit. The *DRD2*rs1799732, *DRD2*rs6277, and *COMTrs4680* genes were identified by real-time PCR. The 48 bp *DRD4*-VNTR and 40 bp *DAT1-*VNTR genes were genotyped using endpoint PCR and subsequent separation of the fragments on a 1.4% agarose gel. Details of reagents and PCR conditions are shown in Appendix A. All polymorphisms were analyzed twice.

The positions of the five polymorphisms are shown in Appendix A. Their genotypes were grouped as follows: (i) *DRD2*rs1799732, only InsIns homozygotes were compared with InsDel heterozygotes (due to the relatively low frequency of the Del allele); (ii) D*RD2rs6277*, the three groups were compared (homozygous C, heterozygote CT, and homozygous T); (iii) *COMTrs4680*, the three groups (Met homozygotes, ValMet heterozygotes, and Val homozygotes) were compared; (iv) *DRD4*-VNTR, genotypes formed by one or two DRD4-L alleles were compared with genotypes formed by two alleles of DRD4-S; (v) *DAT1*-VNTR, nine-repeat homozygotes (9R9R), nine- and ten-repeat heterozygotes (9R10R), and ten-repeat homozygotes (10R10R) were compared.

We used an MLGP score that reflected the total number of the five DA genotypes, in parallel to the general approach used by other authors [41,58,59]. Genotypes purportedly associated with low DA signaling received a score of 0, and those purportedly associated with high DA received a score of 1. In addition, genotypes associated with an intermediate signaling force received a score of 0.5. Specifically, InsIns genotypes in *DRD2*rs1799732, TT in *DRD2*rs6277, ValVal in *COMT*rs4680, 9R9R in *DAT1*-VNTR, and carriers of the DRD4-L variant in *DRD4*-VNTR were assigned a score of 0; InsDel genotypes in *DRD2*rs1799732, CC in *DRD2*rs6277, MetMet in *COMT*rs4680, 10R10R in *DAT1*-VNTR, and carriers of the DRD4-S variant in *DRD4*-VNTR were assigned a score of 1; heterozygotes CT *DRD2*rs6277, ValMet in *COMT*rs4680, and 9R10R in *DAT1*-VNTR received a score of 0.5.

The Shapiro–Wilk test was applied to determine whether these variables followed a normal distribution. The quantitative variables with normal distribution were calculated as the mean and standard deviation (mean ± sd). The qualitative variables were summarized by the absolute and relative frequencies of each of their categories. Comparisons between groups were made via the Student’s *t*-test when the variables were quantitatively normal or via Mann–Whitney U when not. For qualitative variables, the chi-square test with continuity correction or Fisher’s exact test were used. We calculated the OR with a 95% CI.

To determine whether the clinical–demographic variables influenced the statistically significant associations observed among the presence of obesity, other altered cardiometabolic disorder parameters, and SNP genotypes, an analysis was performed using multivariable logistic regression models, including the variables that presented a *p*-value ≤0.15 in univariate analysis [61]. The obesity indicators and the remaining cardiometabolic disorder parameters constituted the dependent variables, while polymorphisms and MLGP score constituted the independent variables. Sex, age, onset, tobacco, alcohol and/or cannabis use, AP treatment type and daily dose, and concomitant treatment were included as covariates. Adjusted OR was calculated with a 95% CI. Nagelkerke’s R^2^ coefficient was used to estimate the approximate percentage of variance explained by the models. The goodness of fit to the model was assessed using the Hosmer–Lemeshow test. Variables with the highest *p*-value were discarded until reaching a model in which all covariates were significant and the goodness of fit to the model was adequate.

The statistical power was calculated following the formulas used In the pwr and Soper DS library (Multiple Regression Post-hoc Statistical Power Calculator et al., 2022). The level of statistical significance was set at 0.05 (Appendix A). All statistical analyses were performed using the statistical program R 4.0.1 and IBM SPSS Statistics for Windows, Version 23.0.

## 3. Results

### 3.1. Sample Characteristics, Obesity Indicators, and Cardiometabolic Parameters

The clinical–demographic features, obesity frequency, and other altered cardiometabolic disorder markers of patients on antipsychotic treatment are shown in Table 1. Three patients had missing data for glucose levels and AP daily doses, two patients had missing data for marital status, alcohol consumption, W, SBP, DBP, TG, and HDL levels, one patient had missing data for diagnosis, age, tobacco use, cannabis, antipsychotic treatment, and height, and 17 patients had missing data for the age of illness onset. Data regarding pharmacological treatments are shown in Table 2. One patient had missing data for treatment. Overall, 31% of patients were treated with clozapine or olanzapine, the two SGAs that present the highest risk of inducing weight gain or worsening of metabolic parameters, whilst more than 50% of patients were treated with medium-risk SGAs, with quetiapine, risperidone, and paliperidone being the most prescribed.

When performing the univariate analysis, we observed some associations between clinical–demographic variables and the cardiometabolic parameters studied. We found that BMI was related to time of illness, and it was higher in patients with fewer years of illness (normal BMI: 23.7 ± 11.5 y, n = 74 and obesity BMI: 19.5 ± 10.7 y, n = 192, *p* = 0.008). When evaluating W, we saw that women had a higher risk of abdominal obesity than men (women/men: 73%/48%; OR: 2.90 [1.70–4.95]). Likewise, we observed that patients with high SBP and high DBP were older (normal SBP: 42.4 ± 10 y, n = 177; high SBP: 46.8 ± 10.6 y, n = 105, *p* = 0.002, and normal DBP: 42.4 ± 10 y, n = 164, high BDP; 45.7 ± 10.7 y, n = 118, *p* = 0.012) and had more time of illness (normal SBP 19.5 ± 10.3 y, n = 166; high SBP: 22.9 ± 12 y, n = 99, *p* = 0.038, and normal BDP: 19.3 ± 10.7 y, n = 157; high BDP; 22.8 ± 11.2 y, n = 108, *p* = 0.013). When evaluating metabolic parameters, we found that the risk of having HTG was related to the onset of the illness, being higher in patients with later onset (normal TG: 22.0 ± 7.7 y, n = 152; HTG: 24.5 ± 8.5 y, n = 113, *p* = 0.028). We also saw that the risk of hyperglycemia (Hglu) was higher in the oldest (normal Glu: 42.6 ± 10.6 y, n = 201; Hglu: 47.1 ± 10.3 y, n = 81, *p* = 0.001) and in those individuals with the longest illness time (normal Glu: 19.4 ± 11 y n = 187; Hglu: 24.2 ± 10.6 y, n = 77, *p* = 0.001). Other parameters did not reach significant associations (Appendix A).

In relation to treatment, we found that patients with Hglu were in treatment with more than one antipsychotic (Hglu: mono/polytherapy 21.7%/39.1%, *p* = 0.002, OR: 2.37 [1.4–4.02]. Likewise, we saw that the risk of arterial hypertension was higher in patients with longer antipsychotic treatment (normal SBP: 25.3 ± 40.2 months, n = 177; high SBP: 31.8 ± 43.0 months, n = 105; *p* = 0.013 and normal BDP: 24.3 ± 38.2 months, n = 164; high BDP: 32.5 ± 45.2 months, n = 118, *p* = 0.047). We did not find significant associations with other parameters (Appendix A).

### 3.2. Relationship between the Presence/Absence of Obesity, BMI, and Abdominal Obesity and the Studied Genotypes

All SNPs were in Hardy Weinberg equilibrium. We observed statistically significant associations for BMI and W according to their genotype in *DRD2*rs1799732, *DAT1-*VNTR, and *DRD4*-VNTR (Figure 1). In relation to BMI, we found that InsIns homozygous patients in *DRD2*rs1799732 had a higher frequency of obesity BMI than patients carrying the Del allele (Figure 1a, X^2^: 8.12, df: 1, *p* = 0.004, OR: 2.76 [1.41–5.40]). In the case of *DAT1*-VNTR, 9R9R homozygotes patients in *DAT1*-VNTR showed a tendency toward a higher risk of obesity BMI than 10R10R homozygotes patients (Figure 1b, X^2^: 4.24, df: 2, *p* = 0.120, OR: 2.82 [1.02–7.82]). These associations remained statistically significant between BMI and *DRD2*rs1799732 and *DAT1*-VNTR when considering clinical–demographic variables (Table 3). The models for *DRD2*rs1799732 and *DAT1*-VNTR explained approximately 8.4% and 6.7% of the variability of the identification accuracy for BMI, respectively (Table 3). In relation to *DRD4*-VNTR, the patients carrying one or two DRD4-L variants in *DRD4*-VNTR had a tendency for a higher risk of obesity BMI than patients carrying two DRD4-S variants In *DRD4*-VNTR (X^2^: 3.54, df: 1, *p* = 0.060, OR: 1.77 [1.01–3.09]). When including the remaining demographic and clinical variables in the model, no relation was confirmed.

Likewise, for W, we found that patients with one or two DRD4-L variants in *DRD4*-VNTR or 9R9R homozygous in *DAT1*-VNTR had a higher risk of abdominal obesity than carriers of two DRD4-S variants (Figure 1c, X^2^: 3.76, df: 1, *p* = 0.052, OR: 1.67 [1.02–2.74]) and 10R10R homozygotes in *DAT1*-VNTR (Figure 1d, X^2^: 6.05, df: 2, *p* = 0.049, OR: 2.77 [1.20–6.36]). The associations found remained significant when including the remaining demographic and clinical variables (*DRD4*-VNTR: X^2^: 20.89, df: 2, *p* < 0.001 and *DAT1*-VNTR: X^2^: 22.12, df: 3, *p* < 0.001) (Table 3). The models explained approximately 9.5% and 10.1% of the variability of the identification accuracy for W, respectively. No associations between *DRD2*rs6277 or *COMT*rs4680 and BMI and W were detected (Appendix A).

Lastly, we found that a shorter illness time with SGA was related to a higher risk of obesity BMI, and that women had a higher risk of presenting abdominal obesity than men (Table 3).

### 3.3. Relationship between the Presence of High Diastolic Blood Pressure, Hyperlipidaemia, and Hyperglycaemia and the Studied Genotypes

We did not observe any association between the cardiometabolic disorder parameters classified according to ATP-III criteria and the genotypes of the *DRD2* and *DRD4* studied genes (Appendix A).

Regarding *DAT1*-VNTR, 9R9R homozygotes patients had a higher frequency of elevated DBP and HTG levels than 10R10R homozygous patients (Figure 2A, DBP: X^2^: 9.82, df: 2, *p* = 0.007 OR: 3.33 [1.54–7.23] and Figure 2B, TG: X^2^: 12.70, df: 2, *p* = 0.002, OR: 3.94 [1.77–8.76]). For *COMT* rs4680, we found that ValMet heterozygous patients showed a tendency to have a lower frequency of HTG levels than ValVal homozygotes (X^2^: 5.87, df = 2, *p* = 0.058, OR: 0.55 [0.32–0.97]). In SBP, serum levels of HDL and glucose parameters exhibited no relationship (Appendix A).

The associations found between DBP and TG levIls in *DAT1*-VNTR polymorphism remained significant when including the remaining demographic and clinical variables in the *DAT1*-VNTR model (Table 4). The models explained approximately 7.5% and 9.9% of the variability of the identification accuracy for DBP and TG, respectively. Furthermore, we observed that older patients had a higher risk of presenting high DBP levels (Table 4). For rs4680, no association with TG was found when the remaining variables were included.

### 3.4. Relationship of the Presence/Absence of Obesity and Cardiometabolic Disorder Parameters Alterations with MLGP Score

Patients with elevated BMI, W, DBP, and TG levels had a lower MLGP score compared with patients who did not present blood pressure and metabolic alterations and obesity (Figure 3). These associations remained statistically significant when considering clinical–demographic variables (Table 5). Thus, in relation to BMI and W, we found that the model explained approximately 7.6% and 9.9% of the variability of the identification accuracy for BMI and W, respectively. In addition, regarding DBP and TG, we found that the model explained approximately 6.6% and 6.3% of the variability of the identification accuracy for DBP and TG, respectively. Any relationship of SBP and serum levels of HDL and glucose with MLPG score was not detected (MLPG score (mean ± sd), normal SBP 2.29 ± 0.87, n = 178 and high SBP: 2.32 ± 0.91, n = 105, *p* = 0.764; normal HDL: 2.36 ± 0.90, n = 131 and low HDL: 2.25 ± 0.86, n = 152, *p* = 0.270; normal Glu: 2.32 ± 0.86, n = 201 and high Glu: 2.27 ± 0.92, n = 81, *p* = 0.669).

## 4. Discussion

Our study showed that three polymorphisms with low DA activity (InsIns in *DRD2*rs1799732, DRD4-L allele in *DRD4*-VNTR, and 9R9R homozygote in *DAT1*-VNTR) were associated with the SGA treatment-induced risk of developing cardiometabolic disorder parameters (obesity BMI, abdominal obesity, high DBP, and HTG) in patients with a severe mental disorder. Importantly, this is the first study using a biologically reported MLGP score in psychiatric patients treated with SGAs, and it shows that the MLGP score was more sensitive for detecting risk of suffering these side effects than each individual genotype. Specifically, a hypofunctional dopaminergic system was associated with an increased risk of obesity, high DBP, and HTG in patients treated with SGAs.

Regarding the *DRD2* gene, a higher obesity BMI was associated with the InsIns genotypes in *DRD2*rs1799732. We did not observe any association of *DRD2*rs1799732 with the presence of abdominal obesity, and with alterations in other cardiometabolic disorder parameters nor between the evaluated parameters and the *DRD2*rs6277. Several studies have analyzed the relationship between genetic variants in rs1799732 and rs6277 and the development of obesity, metabolic alterations, or blood pressure produced by SGA treatment. We did not find any association between obesity and *DRD2* SNPs when the W measurement was used, nor when evaluating the relationship between BMI or W with the *DRD2*rs6277, in accordance with other studies [47,49,51,52,62]. However, an association between the Del allele of the SNP *DRD2*rs1799732 and weight gain in primarily African-American patients with first-episode origin after three months of SGA treatment has been reported [45]. Differences in the demographic characteristics of the samples may determine the occurrence of these associations. In this sense, the genetic variant C-rs6277 allele in SGA-treated chronic schizophrenic patients was found to be associated with greater weight gain only in the African-American origin subpopulation but not in the general group of patients [46]. A similar picture is observed regarding SGA treatment-derived alterations in other cardiometabolic disorder parameters. While a study conducted in 490 Caucasian patients did not detect any association between rs1799732 and two other genetic variants in *DRD2* and lipid alterations [51], in line with our results, another small sample size study of Brazilian patients found an association between low serum HDL levels and the InsIns-rs1799732 genotype of *DRD2* [47]. Therefore, differences in the allele frequencies of rs1799732 and rs6277 described in various ethnic groups and the relatively small sample size of these studies could explain the discrepant results [45,46,47,49,52,54].

Although our study did not detect any association between *DRD4*-VNTR and alterations in cardiometabolic parameters after SGA treatment, it is important to note that, to our knowledge, no study has previously reported such a relationship. Our finding showing that carriers of DRD4-L alleles had a higher risk of abdominal obesity is in line with the previous relationship described between *DRD4*-VNTR and weight gain after antipsychotic treatment [48]. Furthermore, these results are consistent with other studies where both children and adults, carriers of the DRD4-L allele, ate more foods rich in fat than carriers of the DRD4-S genotype, being related to a higher BMI and higher food cravings [17,63,64,65].

The implication of DAT in the development of obesity and alterations in other cardiometabolic disorder parameters tested after antipsychotic treatment has been poorly studied. To date, only one study has evaluated the relationship between *DAT1*-VNTR and weight gain produced by antipsychotic administration and found no association [54]. In this sense, our study is the first to show a higher frequency of BMI obesity, abdominal obesity, and elevated DBP and HTG in individuals with the 9R9R genotype in *DAT1*-VNTR. Our results are in line with other genetic studies reporting higher BMI in 9R9R homozygous individuals of *DAT1-*VNTR than the general population [40]. Furthermore, clinical studies have demonstrated the anti-obesity effect of tesofensine, a DA reuptake inhibitor [66].

Although COMT is less abundantly expressed in the striatum than DAT [67], several genetic studies have highlighted the *COMT* functional SNP rs4680 in the regulation and function of the DA system in this brain region [68,69]. Our study did not demonstrate any association of *COMT*rs4680 with either obesity indicator, nor with the evaluated metabolic alterations, in accordance with a previous study [52]. However, other studies linked the ValVal and the MetMet genotypes in the European [70] and Chinese [55] populations to metabolic alterations after treatment with SGAs.

We carried out a detailed MLGP to group the cumulative impact of functional polymorphisms whose individual effects could be overlooked. In fact, we observed that patients with obesity, high DBP, and HTG showed a lower MLGP score, indicative of low DA signal, than those without alterations. Despite the relatively small sample, these results highlight the potential of this new biological profiling approach to detect obesity risk and alterations in cardiometabolic disorder parameters in patients receiving SGA treatment. Furthermore, since this relationship, as we and others have shown, was not always observed when studying polymorphisms separately, it represents a more sensitive vulnerability index to a given phenotype than the individual genotypes [41,58,59]. Our results are in apparent disagreement with other studies in the general population reporting a correlation among a higher DA signaling in reward regions, overeating, and obesity [41]. However, a hypofunctioning dopamine system and the associated low reward capacity can also lead to overeating and obesity in individuals in an attempt to compensate for such a reward deficit. On the other hand, there are other possible mechanisms or genetic variants of the receptors modulated by antipsychotic drugs, such as serotonin and histamine, and type 4 melanocortin receptors at the hypothalamic level, as well as metabolizing enzymes of these drugs, satiety-inducing hormones, and energy metabolism, that could be involved [71].

Therefore, the finding that individuals carrying genetic variants in *DRD2*, *DRD4*, and *DAT1* and low MLGP score, associated with a hypofunctional DA system, are related to alterations in cardiometabolic parameters linked to obesity reinforce our results and hypothesis. Therefore, patients with a genetic profile associated with a low DA activity in reward circuits will cause the development of obesity and metabolic alterations in the most vulnerable individuals during SGA treatment, which requires further studies.

Our study evaluated a modified MLGP score using common genetic variations of genes that influence dopaminergic transmission. DA binding DRD2 and DRD4, and DA reuptake and catabolism via COMT are critical factors for individual differences in reward processing [41]. However, modified sensitive MLPG scores could benefit from the additive effects of other genes from different systems, such as the serotonergic or histaminergic receptors (HRs), which have been related to several cardiometabolic disorder parameters [71]. In this sense, a positive relationship between different polymorphisms of the serotonin 2A and 2C receptors and/or HR1 have been observed [72], with regard to the obesity and MS development risks in patients who receive antipsychotic drug treatment.

For instance, there are multiple data indicating that HR1 blockade contributes to obesity caused by antipsychotics. In animal models, it has been observed that the deletion of HR1 suppress the weight gain induced by antipsychotics [73], and that the administration of an antagonist, clozapine, can reduce the expression of this receptor in the hypothalamus, correlating with weight gain [74]. In addition, co-treatment with betahistine (HR1 agonist/HR3 antagonist) was reported to be effective in reducing olanzapine-induced weight gain during the chronic phase [75]. Association studies with HR1 gene polymorphisms are scarce, although an interaction has been described between the rs3460704 and rs346070 genetic variants and BMI and obesity in patients treated with antipsychotics drugs with high-affinity for HR1 with those of low affinity [72]. The additive effect of all these genes could contribute to improve the predictive value of the MLGP, which could reduce the variance to less than 10%.

Lastly, when studying the influence of other phenotypes, as in previous studies, we did not observe any influence of sex, age, or the type and time of antipsychotic treatment, AP daily doses, and concomitant treatment on the relationship of weight gain or BMI with the studied genotypes [45,46,47,51]. On the other hand, we observed that BMI was associated with the illness duration, being higher in those patients with fewer years of illness and SGA treatment, agreeing with other studies indicating that the critical period for developing obesity and alterations in other cardiometabolic disorder parameters is within the first 18 months of treatment [14,76,77]. In addition, we found that abdominal obesity was higher in women than in men in accordance with previous studies [78]. Regarding DBP, we showed that DBP levels were related to age, being higher in those patients with older age [79]. However, the relationship between the illness with later age onset and TG levels in psychiatric patients treated with SGA disagrees with previous studies [1,2]. The fact that lipid alterations have been described in the early stages of illness in subjects before receiving antipsychotic treatment, as consequence of unhealthy behaviors associated with life stress, supports our results [4].

It is unclear whether there is a precise association between antipsychotic polypharmacy and higher rates of altered cardiometabolic parameters when compared with patients receiving SGA monotherapy [80,81]. We found that combination of antipsychotic therapy was not independently associated with the prevalence of these alterations. The fact that, in 39% of patients receiving combined therapy with other SGAs, such as aripiprazole, asenapine, or ziprasidone, a low weight gain was associated with metabolic burden [2,82,83] supports our results. As mentioned above, antipsychotic-associated weight gain and alterations in other cardiometabolic disorder parameters are more observable within the first 18 months treatment. In this sense, our study sample represents patients who have been ill for 15 to 25 years, with antipsychotic treatment for at least 27 months; thus, previous treatments in their current regimens could induce partial loss of signal attributable to the beginning of the antipsychotic treatments. Hence, it is important to confirm these findings in psychiatric patients who start treatment with SGAs. Moreover, we must remember that 209 patients (73.6%) demonstrated schizophrenia-like symptoms. Therefore, the use of an MLGP score of cardiometabolic risk could be a useful tool in the management of these patients.

The main strength of our study is its naturalistic character that was performed in patients with several mental disorders undergoing treatment with different drugs. Polypharmacy (antipsychotics, together with antidepressants and/or mood stabilizers, etc.) may influence the development of obesity and metabolic disorders, decreasing the signal attributable to the present SGA treatment. Naturalistic studies are more easily extrapolated to clinical practice since they are closer to therapeutic reality. Another strength is that we used two different indicators of obesity (BMI and W), which refer to different forms of body fat distribution. W refers to the distribution of visceral fat in the abdomen, which is directly related to cardiometabolic alterations. Many studies using anthropometric measures (BMI or weight gain) are notable to distinguish between lean and visceral fat.

The lack of information on several factors, including evaluation of adherence, symptoms, and diet, although they are of high relevance, could not be addressed due to the naturalistic character and cross-sectional nature of our study, thus representing an important limitation. In addition, the characteristics of the sample, constituting Caucasian participants (with less than 5% of different ethnic groups), limit the generalizability of the results. However, the fact that the different ethnic groups in our territory (Bizkaia) is very low, thus decreasing the heterogeneity, provides robustness to our results and interpretation. A longitudinal study with patients from different ethnic groups starting treatment with SGAs to assess its effect on cardiometabolic disorder parameters and to collect data on eating habits and symptoms during AP treatment (i.e., PANSS, YMRS, and MDRS or some equivalent) could be more suitable.

Although our sample size was large in comparison to previous studies, the frequency of some genotypes was low. A limitation of this field is the lack of statistical power to detect true differences between comparison groups. In this respect, we evaluated the MLGP score as an additive effect of the five genotypes. Future studies with greater statistical power checking for or ruling out possible interactions between these genotypes are needed, along with studies determining how these interactions can modulate the response capacity of the reward system.

We proposed the study of candidate genes in DRD2, a target of all antipsychotics. The genes, involved in the DA system, might affect the reward mechanisms of the mesolimbic and mesocortical pathways. However, there are other possible mechanisms or genetic variants of the receptors modulated by antipsychotic drugs, such as serotonin and histamine, and type 4 melanocortin receptors at the hypothalamic level, as well as metabolizing enzymes of these drugs, satiety-inducing hormones, and energy metabolism, that could be involved [71].

## 5. Conclusions

To our knowledge, this is the first exploratory study showing the association of the genetic variants *DRD4-*VNTR and *DAT1-*VNTR, together with an MLGP score analysis, with an increased risk of developing obesity, diastolic arterial hypertension, and hypertriglyceridemia in patients treated with SGAs. Our study consistently demonstrated that a genetic profile associated with a hypofunctional dopaminergic system (low density of DRD2 and DRD4, as well as a reduced DA availability in the synaptic space) would contribute to the pathophysiology underlying the development of alterations in cardiometabolic disorder parameters induced by long-term treatments with SGAs. Furthermore, we showed that the MLGP score is a more reliable index to detect risk side effects than each individual genotype. We propose to perform a longitudinal study with a larger sample size in patients starting treatment with SGAs in order to validate the present conclusion.

## Figures and Tables

**Figure 1 pharmaceutics-15-02134-f001:**
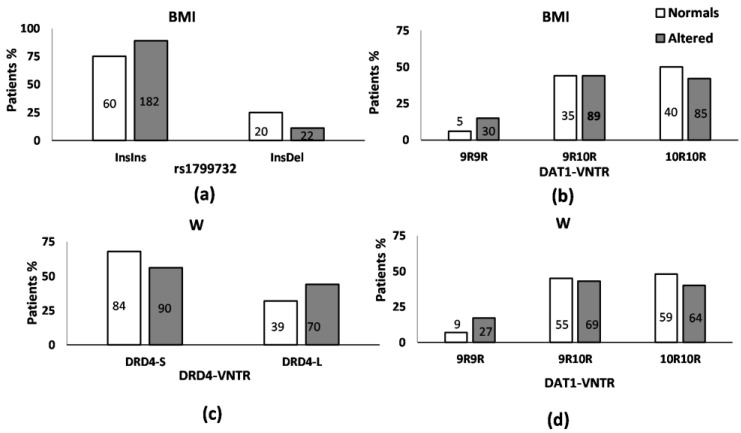
(**a**) Percentage distribution of *DRD2*rs1799732 genotype, according to the presence/absence of obesity, following the WHO criteria for BMI in antipsychotic-treated patients. Association analysis and OR [95% CI] using univariate logistic regression were: InsIns, OR: 2.76 [1.41–5.40]. (**b**) Percentage distribution of *DAT1-*VNTR genotype according to the presence/absence of obesity following the WHO criteria for BMI in antipsychotic-treated patients. Association analysis and OR [95% CI] using univariate logistic regression: 9R9R vs. 10R10R, OR: 2.82 [1.02–7.82]. (**c**) Percentage distribution of *DRD4-*VNTR genotype, according to the presence/absence of obesity, following the ATP-III criteria for waist circumference in antipsychotic-treated patients. Association analysis and OR [95% CI] using univariate logistic regression: DRD4-L, OR: 1.67 [1.02–2.74]. (**d**) Percentage distribution of *DAT1*-VNTR genotype, according to the presence/absence of obesity, following the ATP-III criteria for waist circumference in antipsychotic-treated patients. Association analysis and OR [95% CI] using univariate logistic regression: 9R9R vs. 10R10R, OR: 2.77 [1.20–6.36]. The number of patients in each group is shown inside each bar.

**Figure 2 pharmaceutics-15-02134-f002:**
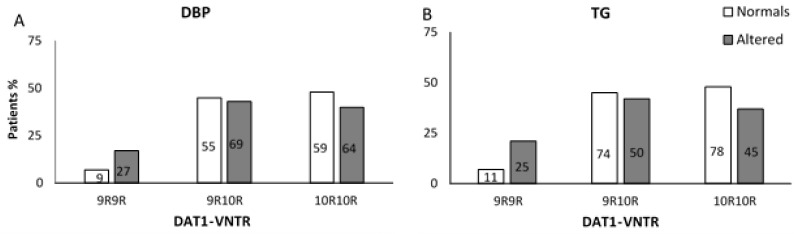
(**A**) Percentage distribution of *DAT1*-VNTR genotypes, according to the presence/absence of altered diastolic blood pressure values. Association analysis and OR [95% CI] using univariate logistic regression: 9R9R vs. 10R10R, OR: 3.33 [1.54–7.23]. (**B**) Percentage distribution of *DAT*1-VNTR genotypes, according to the presence/absence of triglyceride levels in antipsychotic-treated patients. Association analysis and OR [95% CI] using univariate logistic regression: 9R9R vs. 10R10R, OR: 3.94 [1.77–8.76]. The number of patients in each group is shown inside each bar.

**Figure 3 pharmaceutics-15-02134-f003:**
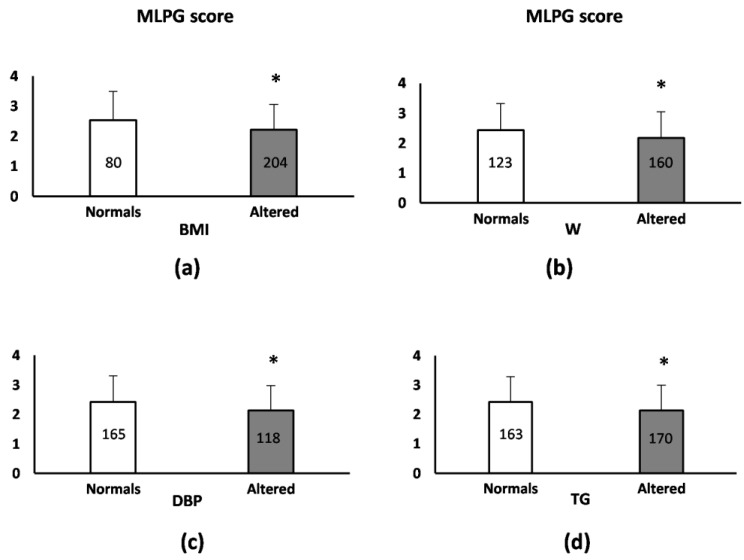
Mean MLPG score according to the presence/absence of obesity following the WHO criteria for BMI (**a**), the ATP-III criteria for waist circumference (**b**), the presence/absence of altered values in diastolic blood pressure (**c**), and triglyceride levels (**d**) in antipsychotic-treated patients. MLGP score data are expressed as the mean ± sd. The number of patients in each group is shown inside each bar. * Significant association according to Mann–Whitney U test: (**a**) *p* = 0.011, (**b**) *p* = 0.016, (**c**) *p* = 0.012, and (**d**) *p* = 0.005.

**Table 1 pharmaceutics-15-02134-t001:** Clinical–demographic data, diagnosis and treatment of the sample.

Variables		N	Mean
**Age** (years)		284	43.8 ±10.7
**Onset** (years)		268	23.0 ± 8.2
**Illness duration** (years)		268	20.7 ± 11.1
**Sex** (men)		188	66.0
**Educational level**	Without studies	15	5.3
	Primary	139	48.8
	High school	98	34.4
	University	32	11.6
**Marital status**	Single	229	80.9
	Married, couple	23	8.1
	Widowed, divorced	31	11.0
**Tobacco use**	User	190	66.9
	Not user	83	29.2
	Former user	11	3.9
**Alcohol use**	Ocassional user	45	15.9
	Not user	231	81.6
	Former user	7	2.5
**Cannabis use**	Ocassional user	40	14.1
	Not user	235	82.7
	Former user	9	3.2
**Diagnosis**	Schizophrenia	209	73.6
	Bipolar disorder	48	16.9
	Psychotic disorder	16	5.6
	Personality disorder	7	2.5
	Others	4	1.5
**AP treatment**	Monotherapy n (%)	169	59.3
	Polytherapy n (%)	115	40.4
	Daily doses (mg/day)	281	669.8 ± 445.7
	Treatment time (months)	284	27.9 ± 41.4
**Concomitant treatment**	Mood stabilizers	69	24.2
	Antidepressants	51	17.9
	Anxiolytics	185	65.1
**Obesity, cardiometabolic disorders parameters**	Obesity, BMI ^a^	204	71.8
	Abdominal obesity, W ^b^	160	56.5
	High SBP ^b^	105	37.1
	High DBP ^b^	118	41.7
	HTG ^b^	120	42.4
	L-HDL ^b^	152	53.7
	Hyperglycemia ^b^	81	28.7

AP: antipsychotic, SBP and DBP: systolic and diastolic blood pressure, HTG: hypertriglyceridemia, L-HDL: low levels of high-density lipoproteins. Diagnostic criteria of WHO ^a^ and ATP-III ^b^.

**Table 2 pharmaceutics-15-02134-t002:** Type of antipsychotic treatment and number of patients treated with 1, 2, or 3 antipsychotics.

Antipsychotic Treatment (AP)		N	%	1AP	2AP	3AP
		284	100	169 (59.5%)	102 (35.9%)	13 (4.6%)
**SGA**	Clozapine	87	30.6	38	43	6
	Olanzapine	75	26.4	45	25	5
	Quetiapine	44	15.5	20	18	6
	Risperidone	45	15.8	21	21	3
	Paliperidone	48	21.1	30	24	4
	Amisulpride	12	4.2	1	9	2
	Aripiprazole	51	17.3	12	32	7
	Asenapide	6	2.1	2	4	0
	Lurasidone	1	0.4	0	0	1
	Fluphenacin	1	0.4	0	1	0
**FGA** combined with SGA	Haloperidol	9	3.2	0	7	2
	Clotiapine	14	4.9	0	11	3
	Zuclophentyxol	7	2.5	0	7	0
	Levopromycin	2	0.7	0	2	0

SGA: second-generation antipsychotics; FGA: first-generation antipsychotics. 1AP, 2AP, 3AP: patients treated with 1, 2, or 3 antipsychotics.

**Table 3 pharmaceutics-15-02134-t003:** Results of the multivariate logistic regression model (MLR) between the presence/absence of obesity, following the WHO criteria for BMI and ATP-III criteria for waist circumference, and the genotypes of the polymorphisms.

MLR			X^2^	Nag R^2^	OR	95% CI	*p*
**BMI**	*DRD2*rs1799732	ref InsDel	15.91	0.084			
	InsIns				2.91	1.42–5.94	**0.003**
	Illness duration				0.97	0.94–0.99	**0.010**
	*DAT1*-VNTR	ref 10R10R	12.61	0.067			
	9R9R				3.23	1.04–10.01	**0.042**
	9R10R				1.13	0.64–2.01	0.671
	Illness duration				0.97	0.94–0.99	**0.006**
**W**	*DAT1*-VNTR	ref 10R10R	22.12	0.101			
	9R9R				2.73	1.16–6.40	**0.021**
	9R10R				1.22	0.73–2.04	0.447
	Sex (Wo/M)				2.88	1.68–4.95	**<0.001**
	*DRD4*-VNTR	ref DRD4-S	20.89	0.095			
	DRD4-L 1 or 2				1.73	1.04–2.87	**0.035**
	Sex (Wo/M)				2.95	1.72–5.06	**<0.001**

VNTR: variable number of tandem repeats; *DRD2*: dopamine receptor type 2; *COMT*: catechol O-methyltransferase; *DAT1*: dopamine transporter; *DRD4*: dopamine receptor type 4; Wo: women; M: men; BMI: body mass index; W: waist circumference; Nag R^2^: Nagelkerke’s R^2^. Significant associations are shown in bold (*p* < 0.05).

**Table 4 pharmaceutics-15-02134-t004:** Results of the multivariate logistic regression model between the presence/absence of altered diastolic blood pressure and triglyceride levels using ATP-III criteria in antipsychotic-treated patients and the genotypes of the studied polymorphisms.

MLR			X^2^	Nag R^2^	OR	95% CI	*p*
**DBP**	*DAT1-*VNTR	ref 10R10R	15.88	0.075			
	9R9R				3.33	**1.54–7.31**	**0.003**
	9R10R				1.36	0.81–2.28	0.252
	Age				1.03	**1.01–1.05**	**0.015**
**TG**	*DAT1*-VNTR	ref 10R10R	20.26	0.099			
	9R9R				4.38	**1.85–10.36**	**0.001**
	9R10R				1.05	0.61–1.79	0.867
	Onset				1.04	**1.01–1.07**	**0.014**

VNTR: variable number of tandem repeats; *DRD2*: dopamine receptor type 2; *COMT*: catechol O-methyltransferase; *DAT1*: dopamine transporter; DBP: diastolic blood pressure; TG: triglyceride; Nag R^2^: Nagelkerke’s R^2^. Significant associations are shown in bold (*p* < 0.05).

**Table 5 pharmaceutics-15-02134-t005:** Results of the multivariate logistic regression model (MLR) between the presence/absence of obesity, altered diastolic blood pressure and triglyceride levels, and the multilocus genetic profile score in antipsychotic-treated patients.

Variable		Univariant Analysis		MLR			
		*p*	X^2^	Nag R^2^	OR	95% CI	*p*
**BMI**			14.38	0.076			
	MLGP ^1^ score	0.011			0.81	0.69–0.95	**0.010**
	Onset	0.066					
	Illness duration	0.008			0.97	0.94–0.99	**0.007**
**W**			21.80	0.099			
	MLGP score	0.016			0.852	0.74–0.97	**0.021**
	Sex (Wo/M)	<0.001			2.92	1.70–4.99	**<0.001**
	Age	0.101					
	Onset	0.086					
	Treatment time	0.124					
**DBP**			14.18	0.066			
	MLGP score	0.012			0.82	0.71–0.94	**0.006**
	Age	0.002			1.03	1.01–1.05	**0.014**
	Illness duration	0.038					
	Treatment time	0.013					
	AP daily doses	0.070					
**TG**			12.77	0.063			
	MLGP score	0.002			0.83	0.72–0.96	**0.014**
	Onset	0.028			1.04	1.01–1.07	**0.015**
	Tobacco use	0.141					
	Treatment time	0.056					

BMI: body mass index; W: waist circumference; DBP: diastolic blood pressure; TG: triglycerides; MLGP: multilocus genetic profile; Wo: women; M: men; Nag R^2^: Nagelkerke’s R^2^. Statistically significant values are shown in bold (*p* < 0.05). ^1^ OR was associated with an increase of 0.5 units in the MLGP variable.

## Data Availability

The data presented in this study are available in the present article and its Appendix A.

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
