# Peer review of "Multilocus Genetic Profile Reflecting Low Dopaminergic Signaling Is Directly Associated with Obesity and Cardiometabolic Disorders Due to Antipsychotic Treatment"

_pharmaceutics, 2023, doi:10.3390/pharmaceutics15082134_

Round 1

Reviewer 1 Report

The authors have submitted a research article regarding an impact of multilocus genetic profile score (MLGP) associated closely to dopaminergic activity (i.e. DRD2, DRD4, DAT, and COMT) on the development of obesity and the cardiometabolic disorders, illustrating a hypothesis suggesting that, despite the availability of the individual genotype, the MLGP score is more sensitive for detecting the risk of obesity, in the aspect of antipsychotic-treated patients. The authors discussed the beneficial availability of the MLGP score which appears to predict side effects such as obesity in patients treated with antipsychotics, resulting in reliable perspectives. This issue is of interest, and impact of their results is strong. My overall concern with the article describing the current available data regarding beneficial availability of the MLGP scores against various individual genotype which are available for evaluation of side effects of antipsychotic, offer something substantial that helps advance our understanding of effective management which draws novel class of effective compounds available in clinic.

Regarding the effect of antipsychotic-induced weight gain, it has been well known the contribution of histamine receptor function in brain to obesity, as the authors might know (for instance, PLoS One. 2014, 9(8):e104160; Psychopharmacology (Berl). 2011, 6(2):257-65). To strengthen authors’ perspectives, the authors are strongly recommended to add a “histaminergic neurotransmission” discussion in detail regarding known H1 and H3 receptor function on the basis of MLGP score in patients, for instance. The positive effects of expected outcomes, if known, may influence largely the authors’ perspective.

Author Response

The authors have submitted a research article regarding an impact of multilocus genetic profile score (MLGP) associated closely to dopaminergic activity (i.e. DRD2, DRD4, DAT, and COMT) on the development of obesity and the cardiometabolic disorders, illustrating a hypothesis suggesting that, despite the availability of the individual genotype, the MLGP score is more sensitive for detecting the risk of obesity, in the aspect of antipsychotic-treated patients. The authors discussed the beneficial availability of the MLGP score which appears to predict side effects such as obesity in patients treated with antipsychotics, resulting in reliable perspectives. This issue is of interest, and impact of their results is strong. My overall concern with the article describing the current available data regarding beneficial availability of the MLGP scores against various individual genotype which are available for evaluation of side effects of antipsychotic, offer something substantial that helps advance our understanding of effective management which draws novel class of effective compounds available in clinic. Author’s comment: The authors are grateful to reviewer comments about general topic of this manuscript. Regarding the effect of antipsychotic-induced weight gain, it has been well known the contribution of histamine receptor function in brain to obesity, as the authors might know (for instance, PLoS One. 2014, 9(8):e104160; Psychopharmacology (Berl). 2011, 6(2):257-65). To strengthen authors’ perspectives, the authors are strongly recommended to add a “histaminergic neurotransmission” discussion in detail regarding known H1 and H3 receptor function on the basis of MLGP score in patients, for instance. The positive effects of expected outcomes, if known, may influence largely the authors’ perspective. Author’s answer to rev. 1: We include a new paragraphs in the Discussion section, page 13, lines 460 to 481, in order to strengthen our perspectives, to add a “histaminergic neurotransmission” discussion in detail regarding known H1 and H3 receptor function on the basis of MLGP score in our patients. Page 13, paragraphs 3&4, lines 460-481: ”…. Our study evaluates a modified MLGP score using common genetic variations of genes that influence dopaminergic transmission. DA binding DRD2 and DRD4, DA reuptake and catabolism via COMT, are critical factors for individual differences in re-ward processing [41]. However, modified sensitive MLPG scores could benefit from the additive effects of other genes from different systems, such as the serotonergic or histaminergic receptors (HR), which it has been related with several cardiometabolic disorder parameters [71]. In this sense, a positive relationship between different polymorphisms of the serotonin 2A and 2C receptors and/or HR1 have been observed [72], in regards with the obesity and MS development risks in patients which receive antipsychotic drug treatment. For instance, there are multiple data indicating that HR1 blockade contributes to obesity caused by antipsychotics. In animal models, it has been observed that the deletion of HR1 suppress the weight gain induced by antipsychotics [73], and that the administration of an antagonist, clozapine, can reduce the expression of this receptor in the hypothalamus, correlating with weight gain [74]. In addition, co-treatment with betahistine (HR1 agonist/HR3 antagonist) has been reported to be effective in reducing olanzapine-induced weight gain during the chronic phase [75]. Association studies with HR1 gene polymorphisms are scarce, although an interaction has been described between the rs3460704 and rs346070 genetic variants, and BMI and obesity in patients treated with antipsychotics drugs with high-affinity for HR1 with those of low affinity [72]. The additive effect of all these genes could contribute to improve the predictive value of the MLGP, which could reduce the variance less than 10%…” Additionally, 4 new references have been included (page 20, lines786-797): 72. Vehof, J.; Risselada, A.J.; Al Hadithy, A.F.Y.; Burger, H.; Snieder, H.; Wilffert, B.; Arends, J.; Wunderink, L.; Knegtering, H.; Wiersma, D.; et al. Association of Genetic Variants of the Histamine H1 and Muscarinic M3 Receptors with BMI and HbA1c Values in Patients on Antipsychotic Medication. Psychopharmacology (Berl.) 2011, 216, 257–265, doi:10.1007/s00213-011-2211-x. 73. Kim, S.F.; Huang, A.S.; Snowman, A.M.; Teuscher, C.; Snyder, S.H. From the Cover: Antipsychotic Drug-Induced Weight Gain Mediated by Histamine H1 Receptor-Linked Activation of Hypothalamic AMP-Kinase. Proc. Natl. Acad. Sci. U. S. A. 2007, 104, 3456–3459, doi:10.1073/pnas.0611417104. 74. Han, M.; Deng, C.; Burne, T.H.J.; Newell, K.A.; Huang, X.-F. Short- and Long-Term Effects of Antipsychotic Drug Treatment on Weight Gain and H1 Receptor Expression. Psychoneuroendocrinology 2008, 33, 569–580, doi:10.1016/j.psyneuen.2008.01.018. 75. Lian, J.; Huang, X.-F.; Pai, N.; Deng, C. Preventing Olanzapine-Induced Weight Gain Using Betahistine: A Study in a Rat Model with Chronic Olanzapine Treatment. PloS One 2014, 9, e104160, doi:10.1371/journal.pone.0104160.

Reviewer 2 Report

The present manuscript if of high interest for the field of pharmacogenetics of psychotropic and antipsychotic treatments.

The approach of a genetic risk score for the implication of the dopamine system for cardiometabolic side effects, in addition to the analysis of the impact of single gene dopamine system polymorphism, is very promising.

Methods and statistical analysis and approaches are described in a very detailed way, supporting the quality of the manuscript.

Study size including 285 adult patients treated by second-generation antipsychotic for at least 3 months is adequate, and the naturalistic study design (treatment prescription in clinical practice without following a study protocol) is of great interest for the clinical application of the results.

Despite inclusion criteria only concerning the type of treatment, three quarter of the study population has been diagnosed with schizophrenia, and thus study results might mainly concern this patient group, which should be mentioned and discussed by the authors.

In addition, the quality of the manuscript might be further improved in discussing risk scores for benefit-risk treatment decisions not only beyond single genetic polymorphisms but also beyond risk scores related to the function of a specific system (e. g. dopaminergic function in this manuscript). Authors already mention a bit the fact that other genetic variants related to efficacy and side effects are concerning additional monoamine systems (e. g. serotonin) as well as transporter and metabolizing enzymes. Nevertheless, they don’t clearly discuss the interest of polygenetic risk score approaches including a much larger variety of gene polymorphisms in several metabolising, transporter as well as action systems.

The interest of such an broader approach should also be discussed in the light of MLGP score (presented by the authors) explaining a significant part of variance in cardiometabolic side effects but still less than 10%.   

For all these reasons, I suggest the acceptance of the manuscript after minor revisions including the points I mentioned above.

Author Response

The present manuscript if of high interest for the field of pharmacogenetics of psychotropic and antipsychotic treatments. The approach of a genetic risk score for the implication of the dopamine system for cardiometabolic side effects, in addition to the analysis of the impact of single gene dopamine system polymorphism, is very promising. Methods and statistical analysis and approaches are described in a very detailed way, supporting the quality of the manuscript. Study size including 285 adult patients treated by second-generation antipsychotic for at least 3 months is adequate, and the naturalistic study design (treatment prescription in clinical practice without following a study protocol) is of great interest for the clinical application of the results. Author’s comment: The authors are grateful to reviewer comments about general topic of this manuscript. Despite inclusion criteria only concerning the type of treatment, three quarter of the study population has been diagnosed with schizophrenia, and thus study results might mainly concern this patient group, which should be mentioned and discussed by the authors. Author’s answer to rev. 2: We include the following sentence in page 14, lines 509-512: “…Moreover, we must remember that 209 patients (73.6%) demonstrated schizophrenia-like symptoms. Therefore, the use of an MLGP score of cardiometabolic risk could be a usefull tool in the management of these patients…” In addition, the quality of the manuscript might be further improved in discussing risk scores for benefit-risk treatment decisions not only beyond single genetic polymorphisms but also beyond risk scores related to the function of a specific system (e. g. dopaminergic function in this manuscript). Authors already mention a bit the fact that other genetic variants related to efficacy and side effects are concerning additional monoamine systems (e. g. serotonin) as well as transporter and metabolizing enzymes. Nevertheless, they don’t clearly discuss the interest of polygenetic risk score approaches including a much larger variety of gene polymorphisms in several metabolising, transporter as well as action systems. The interest of such an broader approach should also be discussed in the light of MLGP score (presented by the authors) explaining a significant part of variance in cardiometabolic side effects but still less than 10%. Author’s answer to rev. 2: We include a new paragraphs in the Discussion section, page 13, lines 460 to 481, in order to strengthen our perspectives, to add a “histaminergic neurotransmission” discussion in detail regarding known H1 and H3 receptor function on the basis of MLGP score in our patients. Page 13, paragraphs 3&4, lines 460-481: ”…. Our study evaluates a modified MLGP score using common genetic variations of genes that influence dopaminergic transmission. DA binding DRD2 and DRD4, DA reuptake and catabolism via COMT, are critical factors for individual differences in re-ward processing [41]. However, modified sensitive MLPG scores could benefit from the additive effects of other genes from different systems, such as the serotonergic or histaminergic receptors (HR), which it has been related with several cardiometabolic disorder parameters [71]. In this sense, a positive relationship between different polymorphisms of the serotonin 2A and 2C receptors and/or HR1 have been observed [72], in regards with the obesity and MS development risks in patients which receive antipsychotic drug treatment. For instance, there are multiple data indicating that HR1 blockade contributes to obesity caused by antipsychotics. In animal models, it has been observed that the deletion of HR1 suppress the weight gain induced by antipsychotics [73], and that the administration of an antagonist, clozapine, can reduce the expression of this receptor in the hypothalamus, correlating with weight gain [74]. In addition, co-treatment with betahistine (HR1 agonist/HR3 antagonist) has been reported to be effective in reducing olanzapine-induced weight gain during the chronic phase [75]. Association studies with HR1 gene polymorphisms are scarce, although an interaction has been described between the rs3460704 and rs346070 genetic variants, and BMI and obesity in patients treated with antipsychotics drugs with high-affinity for HR1 with those of low affinity [72]. The additive effect of all these genes could contribute to improve the predictive value of the MLGP, which could reduce the variance less than 10%…” For all these reasons, I suggest the acceptance of the manuscript after minor revisions including the points I mentioned above. Author’s comment: The authors are grateful to reviewer comments about general topic of this manuscript.

Reviewer 3 Report

Dear All,

Personalization of Treatment is  corner stone of precision medicine. Accordingly the authors do a nice job! The manuscript is very well written, and the introduction provides a good, generalized background of the topic that quickly gives the reader an appreciation of the applications of the technique. Moreover, the topic and the data are very interesting. Certainly, this study will contribute much to the literature.

Note; it would be interesting to combine these results  with pharmacokinetic parameters , i.e.. Dug metabolizing enzymes genotypes

Thank you

Kind Regards

Author Response

Personalization of Treatment is corner stone of precision medicine. Accordingly the authors do a nice job! The manuscript is very well written, and the introduction provides a good, generalized background of the topic that quickly gives the reader an appreciation of the applications of the technique. Moreover, the topic and the data are very interesting. Certainly, this study will contribute much to the literature. Author’s comment: The authors are grateful to reviewer comments about general topic of this manuscript. Note; it would be interesting to combine these results with pharmacokinetic parameters , i.e.. Dug metabolizing enzymes genotypes Author’s comment: We agree with the reviewer. It is of interest to be performed in future research, including these type of data.

Round 2

Reviewer 1 Report

The authors have done a good job responding to reviewer comments and concerns in their revision. I believe the manuscript is significantly improved as a result. Now I recommend that this revised version of the manuscript can be accepted for publication in Pharmaceutics.